# Entanglement gap, corners, and symmetry breaking

Vincenzo Alba[1*]

**1** Institute for Theoretical Physics, Universiteit van Amsterdam, Science Park 904,
Postbus 94485, 1098 XH Amsterdam, The Netherlands
* v.alba@uva.nl

October 17, 2020

## Abstract

We investigate the finite-size scaling of the lowest entanglement gap $\delta\xi$ in the ordered phase of the two-dimensional quantum spherical model (QSM). The entanglement gap decays as $\delta\xi = \Omega/\sqrt{L \ln(L)}$. This is in contrast with the purely logarithmic behaviour as $\delta\xi = \pi^2/\ln(L)$ at the critical point. The faster decay in the ordered phase reflects the presence of magnetic order. We analytically determine the constant $\Omega$, which depends on the low-energy part of the model dispersion and on the geometry of the bipartition. In particular, we are able to compute the corner contribution to $\Omega$, at least for the case of a square corner.

## 1 Introduction

In recent years the cross-fertilization between condensed matter and quantum information fueled an impressive progress in our understanding of quantum many-body systems [1–4]. The entanglement spectrum (ES) has been the subject of intense investigation. Let us consider a system in its ground state $|\Psi\rangle$ and a spatial bipartition of it as $A \cup \bar{A}$ (see Fig. 1). The reduced density matrix $\rho_A = \mathrm{Tr}_{\bar{A}}|\Psi\rangle\langle\Psi|$ of $A$ can be written as

$$\rho_A = e^{-\mathcal{H}_A}. \tag{1}$$

Here $\mathcal{H}_A$ is the so-called entanglement hamiltonian. The entanglement spectrum levels $\xi_i = -\ln(\lambda_i)$, with $\lambda_i$ the eigenvalues of $\rho_A$, are the "energies" of $\mathcal{H}_A$. Early works [5–7] on entanglement spectra aimed at understanding the effectiveness of the density matrix renormalisation group (DMRG) [8, 9] to simulate one-dimensional systems.

Recently, an intense theoretical activity has been devoted to understand the ES in fractional quantum Hall systems [10–21], topologically ordered systems [22–24], magnetically ordered systems [21, 25–37], Conformal Field Theories (CFTs) [38–41], and systems with impurities [42]. The entanglement gap (or Schmidt gap) $\delta\xi$ emerged as a natural quantity to investigate. $\delta\xi$ is the gap of the entanglement hamiltonian, and it is defined as

$$\delta\xi = \xi_1 - \xi_0, \tag{2}$$

where $\xi_0$ and $\xi_1$ are the first two low-laying ES levels. For the standard *energy* gap, i.e., the gap of the physical hamiltonian, there exists a "universal" correspondence between its scaling behaviour and ground state properties, such as the decay of correlation functions [43]. Much less is known for the entanglement gap, although several results are available. For instance, its behaviour at one-dimensional quantum critical points has been investigated [5, 6, 11, 27, 28, 32, 44–46]. In CFTs it is well established that $\delta\xi$ decays logarithmically as $\delta\xi \propto 1/\ln(\ell)$ with $\ell$ the subsystem length [38]. Similar scaling is found in

models that are solvable via the corner transfer matrix technique [44]. Higher-dimensional models are uncharted territory. Interestingly, some explicit counterexamples show that the closure of the entanglement gap in general does not signal criticality [21], also for the momentum-space ES [47]. The scenario is different deep in ordered phases of matter. For instance, the lower part of the ES of magnetically-ordered ground states that break a continuous symmetry [29] is reminiscent of the Anderson tower-of-states [48–50]. This has been verified in systems of quantum rotors [29], in the two-dimensional Bose-Hubbard model in the superfluid phase [31] (see also [37]), and also in Heisenberg antiferromagnets on the square [34] and on the kagome lattice [36]. In the tower-of-states scenario gaps in the lower part of the ES decay as a power-law with the subsystem volume, with multiplicative logarithmic corrections [29]. Higher ES gaps exhibit a slower decay [29, 31, 35].

Given the lack of general results, exactly solvable models can provide valuable insights into the generic features of the entanglement gap. Here we investigate the entanglement gap in the ordered phase of the two-dimensional quantum spherical model (QSM) [51–55]. Despite its appealing simplicity, the QSM contains several salient features of generic quantum many-body systems. The model is mappable to a system of free bosons with an external constraint, implying that its properties can be studied with moderate cost. Its classical version proved to be valuable to validate the theory of critical phenomena and finite size scaling [56]. The ground-state phase diagram of the two-dimensional QSM exhibits a paramagnetic (disordered) phase and a ferromagnetic (ordered) one, which are divided by a continuous quantum phase transition. The universality class is that of the three-dimensional classical $O(N)$ vector model [57] in the large $N$ limit [52, 53, 58]. Entanglement properties of $O(N)$ models have been addressed in the past [59, 60] (see also [61–64] for recent studies in the QSM). Here we consider a two-dimensional lattice of linear size $L$. The typical bipartitions that we use are reported in Fig. 1. Figure 1 (a) shows a bipartition with a straight boundary. The bipartition in Fig. 1 (b) contains a square corner. The effects of corners in the scaling of entanglement-related quantities is nontrivial, and it has been studied intensely in the last decade [4, 65–72].

Our main result is that in the ordered phase of the QSM, in the limit $L, \ell_x, \ell_y \to \infty$ with the ratios $\omega_{x,y} = \ell_{x,y}/L$ (see Fig. 1) fixed the entanglement gap decays as

$$\delta\xi = \frac{\Omega}{\sqrt{L\ln(L)}} + \dots \tag{3}$$

Here the dots denote subleading terms that we neglect. The constant $\Omega$, which we determine analytically, depends on the low-energy properties of the model and on the geometry of the bipartition. In particular, we analytically determine the corner contribution to $\Omega$. The "fast", i.e., power-law behaviour as $1/\sqrt{L}$ in (3) reflects the presence of magnetic order, whereas the logarithmic correction is similar to the critical behaviour [64] of $\delta\xi$. Finally, we should mention that Eq. (3) is different from the result derived in Ref. [29], where it was shown that for $O(N)$ models $\delta\xi \propto (L\ln(L))^{-1}$.

The manuscript is organised as follows. In section 2 we introduce the QSM. In section 3 we review the finite-size scaling of the ground-state two-point correlation functions. In section 4 we briefly overview the calculation of the entanglement gap in the QSM. Section 5 is devoted to the derivation of our main results. In section 6 we provide numerical checks. We conclude in section 7. In A we report some technical derivations.

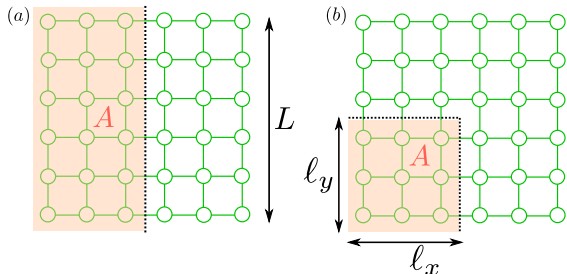

Figure 1: Bipartition of the system as $A \cup \bar{A}$ used in this work. The lattice has $L \times L$ sites and periodic boundary conditions in both directions are used. (a) Bipartition with smooth boundary. (b) Bipartition with a corner. We define the aspect ratios $\omega_x = \ell_x/L$ and $\omega_y = \ell_y/L$.

## 2  Quantum Spherical Model

The quantum spherical model [52–54] (QSM) on a two dimensional cubic lattice of volume $V = L^2$, with $L$ being the lattice linear size, is defined by the hamiltonian

$$H = \frac{g}{2}\sum_{\boldsymbol{n}} p^2 - J\sum_{\langle \boldsymbol{n},\boldsymbol{m}\rangle} s_{\boldsymbol{n}}s_{\boldsymbol{m}} + (\mu - 2)\sum_{\boldsymbol{n}} s_{\boldsymbol{n}}^2. \tag{4}$$

Here, $n = (n_x, n_y)$ denotes a generic lattice site, and $\langle n, m\rangle$ a lattice bond joining two nearest-neighbour sites. We set $J = 1$ in (4). The spin $s_i$ and momenta $p_i$ variables satisfy standard bosonic commutation relations

$$[p_{\boldsymbol{n}}, p_{\boldsymbol{m}}] = [s_{\boldsymbol{n}}, s_{\boldsymbol{m}}] = 0, \quad [s_{\boldsymbol{n}}, p_{\boldsymbol{m}}] = \mathrm{i}\delta_{\boldsymbol{nm}}. \tag{5}$$

Here we refer to the parameter $g$ as the *quantum coupling*. Indeed, in the limit $g \to 0$ the model reduces to the classical spherical model [73, 74]. The spherical parameter $\mu$ is a Lagrange multiplier that fixes the global magnetization as

$$\sum_{\boldsymbol{n}}\langle s_{\boldsymbol{n}}^2\rangle = V, \tag{6}$$

To diagonalize the QSM hamiltonian (4), one can exploit its translational invariance. First, one performs a Fourier transform as

$$p_{\boldsymbol{n}} = \frac{1}{\sqrt{V}}\sum_{\boldsymbol{k}} e^{-\mathrm{i}\boldsymbol{nk}}\pi_{\boldsymbol{k}}\ , \qquad s_{\boldsymbol{n}} = \frac{1}{\sqrt{V}}\sum_{k} e^{\mathrm{i}\boldsymbol{nk}}q_{\boldsymbol{k}}, \tag{7}$$

where the sum is over $\boldsymbol{k} = (k_x, k_y)$ in the first Brillouin zone $k_i = 2\pi/Lj$, with $j \in [-L/2, L/2]$ integer. In Fourier space one obtains

$$H = \sum_{\boldsymbol{k}} \frac{g}{2}\pi_{\boldsymbol{k}}\pi_{-\boldsymbol{k}} + \Lambda_{\boldsymbol{k}}^2\, q_{\boldsymbol{k}}q_{-\boldsymbol{k}}. \tag{8}$$

The single-particle dispersion relation is given as

$$\Lambda_{\boldsymbol{k}} = \sqrt{\mu + \omega_{\boldsymbol{k}}} \quad \text{with} \quad \omega_{\boldsymbol{k}} = 2 - \cos k_x - \cos k_y \tag{9}$$

To fully diagonalise (8) we introduce the new bosonic ladder operators $b_{\boldsymbol{k}}$ and $b_{\boldsymbol{k}}^\dagger$ as

$$q_{\boldsymbol{k}} = \alpha_{\boldsymbol{k}}\frac{b_{\boldsymbol{k}} + b_{-\boldsymbol{k}}^\dagger}{\sqrt{2}}\ , \qquad \pi_{\boldsymbol{k}} = \frac{\mathrm{i}}{\alpha_{\boldsymbol{k}}}\frac{b_{\boldsymbol{k}}^\dagger - b_{-\boldsymbol{k}}}{\sqrt{2}}, \tag{10}$$

where $\alpha_{\boldsymbol{k}}^2 = \sqrt{g/2}\Lambda_{\boldsymbol{k}}^{-1}$. Now, the hamiltonian (8) is fully diagonal, and it reads as

$$H = \sum_{\boldsymbol{k}} E_{\boldsymbol{k}}(b_{\boldsymbol{k}}^\dagger b_{\boldsymbol{k}} + 1/2), \text{ with } E_{\boldsymbol{k}} = \sqrt{2g}\Lambda_{\boldsymbol{k}}. \tag{11}$$

For the following, it is useful to consider the ground-state two-point correlation functions $\langle s_{\boldsymbol{n}} s_{\boldsymbol{m}} \rangle$ and $\langle p_{\boldsymbol{n}} p_{\boldsymbol{m}} \rangle$. They are given as [54]

$$\mathbb{S}_{\boldsymbol{nm}} = \langle s_{\boldsymbol{n}} s_{\boldsymbol{m}} \rangle = \frac{1}{2V} \sum_{\boldsymbol{k}} e^{\mathrm{i}(\boldsymbol{n}-\boldsymbol{m})\cdot\boldsymbol{k}} \alpha_{\boldsymbol{k}}^2 \tag{12}$$

$$\mathbb{P}_{\boldsymbol{nm}} = \langle p_{\boldsymbol{n}} p_{\boldsymbol{m}} \rangle = \frac{1}{2V} \sum_{\boldsymbol{k}} e^{-\mathrm{i}(\boldsymbol{n}-\boldsymbol{m})\cdot\boldsymbol{k}} \alpha_{\boldsymbol{k}}^{-2} \tag{13}$$

$$\mathbb{K}_{\boldsymbol{nm}} = \langle s_{\boldsymbol{n}} p_{\boldsymbol{m}} \rangle = \frac{\mathrm{i}}{2}\delta_{nm} \tag{14}$$

Importantly, the trivial identity holds

$$\mathbb{P}_{\boldsymbol{nm}} = \frac{1}{g} \int d\mu \, \mathbb{S}_{\boldsymbol{nm}}. \tag{15}$$

By using (12), the spherical constraint (6) can be rewritten as

$$\frac{2}{g} = \frac{1}{V}\sum_{\boldsymbol{k}} \frac{1}{E_{\boldsymbol{k}}} = \frac{2}{g}\mathbb{S}_{\boldsymbol{nn}}. \tag{16}$$

Eq. (16) is the so-called gap equation in the context of the large-$N$ model [75]. A crucial observation is that the correlator (12) exhibits a singularity for $\boldsymbol{k} = 0$. This zero mode will play a crucial role in the behaviour of the entanglement gap.

In two dimensions at zero temperature the QSM exhibits a second-order phase transition at a critical value $g_c$. The value of $g_c$ is known analytically as

$$g_c = \frac{\pi^4}{2}K^{-4}\left(1/2 - 1/\sqrt{2}\right) \simeq 9.67826. \tag{17}$$

For $g < g_c$ the QSM exhibits a magnetically ordered phase, which is the focus of this work. At $g > g_c$ the ground state is paramagnetic. Different phases are associated with different behaviour of the spherical parameter $\mu$. In the paramagnetic phase one has that $\mu$ is finite nonzero. On the other hand, $\mu = 0$ at critical point, and in the ordered phase. The different phases of the model correspond to different finite-size scaling behaviours of $\mu$. In the paramagnetic phase one has $\mu = \mathcal{O}(1)$. At the critical point one can show that $\mu = \mathcal{O}(1/L^2)$. In the ordered phase $\mu = \mathcal{O}(1/L^4)$. The critical behaviour at $g_c$ is in the universality class of the three-dimensional $N$-vector model [53] at large $N$.

## 3 Spin and momentum correlators

Here we summarise the finite-size scaling of the spin-spin correlation function $\mathbb{S}_{\boldsymbol{nm}}$ (cf. (12)) and the momentum correlation function $\mathbb{P}_{\boldsymbol{nm}}$ (cf. (13)). Let us focus first on the spin correlator. We are interested in the limit $L \to \infty$. We can decompose the correlator as

$$\mathbb{S}_{\boldsymbol{nm}} = \mathbb{S}_{\boldsymbol{nm}}^{(th)} + \mathbb{S}_{\boldsymbol{nm}}^{(L)} + \dots \tag{18}$$

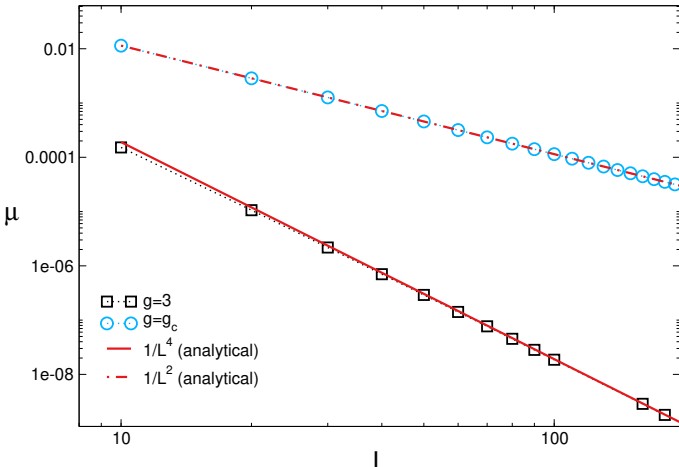

Figure 2: Spherical parameter $\mu$ at the critical point at $g_c \simeq 9.67$ and in the ordered phase at $g = 3$. Symbols are exact numerical results. The lines are analytical results in the large $L$ limit.

The first term is the leading term in the large $L$ limit. Note that the first term depends on $L$ because $\mu$ depends on $L$. The second term in (18) is the first subleading correction in powers of $1/L$. The dots denote more subleading terms that we neglect. The thermodynamic contribution is given as

$$\mathbb{S}_{nm}^{(th)} = \frac{\sqrt{g_c}}{2\sqrt{2}(2\pi)^2} \int d\boldsymbol{k} \frac{e^{i\boldsymbol{k}(\boldsymbol{n}-\boldsymbol{m})}}{\sqrt{\mu + \omega_{\boldsymbol{k}}}}, \tag{19}$$

The finite-size part has the surprisingly simple form [64] as

$$\mathbb{S}_{nm}^{(L)} = \frac{\sqrt{g}}{4\pi} \sideset{}{'}\sum_{l,l'=-\infty}^{\infty} \frac{e^{-\sqrt{2\mu} F_{ll'}(\boldsymbol{n},\boldsymbol{m})}}{F_{ll'}(\boldsymbol{n},\boldsymbol{m})}. \tag{20}$$

Here we defined

$$F_{ll'}(\boldsymbol{n},\boldsymbol{m}) = \sqrt{(lL + n_x - m_x)^2 + (l'L + n_y - m_y)^2}. \tag{21}$$

The prime in the sum means that one has to remove the term with $(l, l') = (0, 0)$. Eq. (20) holds in the limit $L \to \infty$ and $\mu \to 0$, i.e., for $g \leq g_c$. The correlators $\mathbb{S}_{nm}$ depend only on $n_x - m_x$ and $n_y - m_y$, reflecting translation invariance. Moreover, the infinite sums in $l, l'$ enforces that $\mathbb{S}_{nm}$ is periodic along the two directions, i.e., it is invariant under $n_y - m_y \to n_y - m_y \pm L$ and $n_x - m_x \to n_x - m_x \pm L$. Interestingly, $\mathbb{S}_{nm}^{(L)}$ is singular if either $\omega_y = 1$ or $\omega_x = 1$ (see Fig. 1 (a)), whereas no singularity occurs for $\omega_x < 1$ and $\omega_y < 1$, i.e, in the presence of a bipartition with a corner (see Fig. 1 (b)). Let us consider the case $\omega_y = 1$. Now the terms with $l = 0$ and $l' = \pm 1$ in (20) are singular in the limit $n_x - m_x \to 0$ and $n_y - m_y \to \pm 1$. Terms with $|l'| > 1$ or $|l| > 1$ in (20) are not singular, and do not affect the singularity structure of $\mathbb{S}_{nm}$. These singularity will play an important role in section 5.

Similar to (18), we can decompose the momentum correlator as

$$\mathbb{P}_{nm} = \mathbb{P}_{nm}^{(th)} + \mathbb{P}_{nm}^{(L)} + \dots \tag{22}$$

Here we defined

$$\mathbb{P}_{nm}^{(th)} = \frac{1}{4\sqrt{2g}\pi^2} \int_{-\pi}^{\pi} d\boldsymbol{k} e^{i\boldsymbol{k}(\boldsymbol{n}-\boldsymbol{m})} \sqrt{\mu + \omega_{\boldsymbol{k}}}. \tag{23}$$

The finite-size part $\mathbb{P}_{nm}^{(L)}$ has the same structure as (20), and it reads as

$$\mathbb{P}_{nm}^{(L)} = -\frac{1}{4\pi\sqrt{g}} \sideset{}{'}\sum_{l,l'=-\infty}^{\infty} e^{-\sqrt{2\mu}F_{ll'}(\boldsymbol{n},\boldsymbol{m})} \left[ \frac{1}{F_{ll'}^3(\boldsymbol{n},\boldsymbol{m})} + \frac{\sqrt{2\mu}}{F_{ll'}^2(\boldsymbol{n},\boldsymbol{m})} \right], \tag{24}$$

with $F_{ll'}(\boldsymbol{n},\boldsymbol{m})$ as defined in (21). Eq. (23) and Eq. (24) are obtained from Eq. (19) and Eq. (20) by using (15). As for Eq. (20), the finite-size term (24) is singular if subsystem $A$ spans the full lattice in one of the two directions, i.e., if $\omega_y = 1$ or $\omega_y = 1$ (see Fig. 1). For $\omega_y = 1$ the singularity occurs for $l = 0$ and $l' = \pm 1$ in the limit $n_x - m_x \to 0$ and $n_y - m_y \to \pm 1$. Finally, the first term has a stronger singularity than the second one.

## 3.1 Spherical parameter

Let us discuss the finite-size scaling of the spherical constraint $\mu$ (cf. (16)) in the ordered phase of the QSM. For $g \le g_c$ the spherical parameter vanishes in the thermodynamic limit. At the critical point one has the behaviour [64] $\mu \propto \gamma_2^2/(2L^2)$, with $\gamma_2$ a universal constant. To derive the behaviour of $\mu$ in the ordered phase we use Eq. (12) in the gap equation (16). We obtain

$$\frac{1}{\sqrt{g}} = \frac{1}{2\sqrt{2}(2\pi)^2} \int \frac{d\boldsymbol{k}}{\sqrt{\omega_{\boldsymbol{k}}}} - \frac{\sqrt{\mu}}{2\sqrt{2}\pi} + \frac{1}{4\pi L} \sideset{}{'}\sum_{l,l'=-\infty}^{\infty} \frac{e^{-\sqrt{2\mu}L\sqrt{l^2+l'^2}}}{\sqrt{l^2 + l'^2}} \tag{25}$$

As it clear from the exponent in the last term in (25) the scaling as $\mu \propto 1/L^2$ at criticality implies that terms with large $l, l'$ are exponentially suppressed. On the other hand, for $\mu \propto 1/L^4$ this is not the case because the term in the exponent in (25) is $\mathcal{O}(1/L)$. First, we anticipate that the second term in (25) is $\mathcal{O}(1/L^2)$, and it is subleading. To extract the leading behaviour of $\mu$ we use the very elegant identity involving the function $\mathcal{K}(\sigma, d, y)$ defined as [76]

$$\mathcal{K}(\sigma, d, y) = \sideset{}{'}\sum_{\boldsymbol{l}(d)} \frac{K_\sigma(2y|\boldsymbol{l}|)}{(y|\boldsymbol{l}|)^\sigma}, \quad |\boldsymbol{l}| = (l_1^2 + l_2^2 + \cdots + l_d^2)^{\frac{1}{2}}. \tag{26}$$

Here the sum is over the $d$-dimensional vector of integers $l_i \in (-\infty, \infty)$, $K_\sigma(z)$ is the modified Bessel function of the second kind, and $y > 0$ and $\sigma$ are real parameters. We are interested in the case $d = 2$ and $\sigma = 1/2$ (cf. (25)). One can show that [76]

$$\mathcal{K} = \frac{1}{2}\pi^{\frac{d}{2}}\Gamma\left(\frac{d}{2} - \sigma\right)y^{-d} + \frac{1}{2}\pi^{2\sigma - \frac{d}{2}}C(\sigma, d)y^{-2\sigma} - \frac{1}{2}\Gamma(-\sigma)$$
$$+ \frac{1}{2}\pi^{2\sigma - \frac{d}{2}}\Gamma\left(\frac{d}{2} - \sigma\right)y^{-2\sigma}\sideset{}{'}\sum_{\boldsymbol{l}(d)} \left[ \left(|\boldsymbol{l}|^2 + \frac{y^2}{\pi^2}\right)^{\sigma - \frac{d}{2}} - |\boldsymbol{l}|^{2\sigma - d} \right]. \tag{27}$$

The constant $C(\sigma, d)$ for $d = 2$ reads as

$$C(\sigma, 2) = 4\Gamma(1 - \sigma)\zeta(1 - \sigma)\beta(1 - \sigma), \tag{28}$$

where $\zeta(x)$ is the Riemann zeta function, and $\beta(x)$ is the analytic continuation of the Dirichlet series

$$\beta(x) = \sum_{l=0}^{\infty} \frac{(-1)^l}{(2l+1)^x}. \tag{29}$$

To apply (27) we fix $y = \sqrt{\mu/2}L$. In the limit $\mu \to 0$ the leading behaviour of $\mathcal{K}$ is given by the first term on the right hand side in (27). After using that in (25) we obtain

$$\frac{1}{\sqrt{g}} = \frac{1}{8\pi^2\sqrt{2}} \int \frac{d\boldsymbol{k}}{\sqrt{\omega_{\boldsymbol{k}}}} + \frac{1}{2\sqrt{2\mu}L^2}. \tag{30}$$

In (30) we are neglecting vanishing terms in the limit $L \to \infty$. The second term in (30) is also simply obtained by isolating the term with $\boldsymbol{k} = 0$, i.e., the zero mode, in the sum in (16). It is now clear that we can parametrize $\mu$ as

$$\mu = \frac{\gamma_4^2}{L^4}. \tag{31}$$

After substituting in (30), we obtain that

$$\gamma_4 = \left[ \frac{2\sqrt{2}}{\sqrt{g}} - \frac{1}{4\pi^2} \int \frac{d\boldsymbol{k}}{\sqrt{\omega_{\boldsymbol{k}}}} \right]^{-1} \tag{32}$$

Note that the constant $\gamma_4$ is not universal, as it is clear from the explicit dependence on $g$. This is expected, and it is in contrast with the result at the critical point, where $\mu = \gamma_2^2/(2L^2)$, with $\gamma_2$ universal.

## 4 Entanglement gap in the QSM

Here we briefly review how to calculate the entanglement gap in the QSM. Entanglement properties of the QSM are derived from the two-point correlation functions (12) and (13) because the model can be mapped to free bosons (see Ref. [7] for a review). We first define the correlation matrix $\mathbb{C}$ restricted to subsystem $A$ as

$$\mathbb{C}_A = \mathbb{S}_A \cdot \mathbb{P}_A, \tag{33}$$

with $\mathbb{S}_A$ and $\mathbb{P}_A$ defined in (12) and (13), with $\boldsymbol{n}, \boldsymbol{m} \in A$. Since the QSM is mapped to free-bosons, the reduced density matrix of a subsystem $A$ is a quadratic operator, and it is written as [7]

$$\rho_A = Z^{-1} e^{-\mathcal{H}_A}, \quad \mathcal{H}_A = \sum_k \epsilon_k b_k^\dagger b_k. \tag{34}$$

Here $\mathcal{H}_A$ is the so-called entanglement hamiltonian, $\epsilon_k$ are *single-particle* entanglement spectrum levels, and $b_k$ are free-bosonic operators. $Z$ is a normalization factor. The eigenvalues $e_k$ of $\mathbb{C}_A$ are obtained from the $\epsilon_k$ as

$$\sqrt{e_k} = \frac{1}{2} \coth\left(\frac{\epsilon_k}{2}\right). \tag{35}$$

The entanglement spectrum, i.e., the spectrum of $\mathcal{H}_A$ is obtained by filling in all the possible ways the single-particle levels $\epsilon_k$. The lowest ES level is the vacuum state. Thus, the lowest entanglement gap $\delta\xi$ (Schmidt gap) is

$$\delta\xi = \epsilon_1, \tag{36}$$

with $\epsilon_1$ the smallest single-particle ES level, or equivalently, the largest $e_1$ (cf. Eq. (35)).

# 5 Scaling of the entanglement gap in the ordered phase of the QSM

In this section we investigate the scaling of the entanglement gap for $g < g_c$, i.e., in the ordered phase of the QSM. First, it has been numerically observed in Ref. [64] that for $g < g_c$, in the limit $L \to \infty$ the flat vector $|\mathbf{1}\rangle$ defined as

$$|\mathbf{1}\rangle = \frac{1}{\sqrt{|A|}}(1, 1, \ldots, 1), \tag{37}$$

with $|A| = \ell_x \ell_y$, is the right eigenvector of $\mathbb{C}_A$ corresponding to the largest eigenvalue $e_1$, i.e., the zero-mode eigenvector. Moreover, $|\mathbf{1}\rangle$ is also eigenvector of the matrix $\mathbb{S}_A$. It is interesting to investigate the structure of the associated eigenvalue. This is calculated as

$$\langle\mathbf{1}|\mathbb{S}|\mathbf{1}\rangle = \frac{1}{|A|} \sum_{\mathbf{n},\mathbf{m}\in A} \mathbb{S}_{\mathbf{n}\mathbf{m}}. \tag{38}$$

After using (18), it is straightforward to numerically check that the thermodynamic part of the correlator $\mathbb{S}^{(th)}$ for large $L$ gives a subleading term as $L\ln(L)$ in (38) (see section 6). The leading contribution is given by the finite-size part of the correlator $\mathbb{S}^{(L)}$, and it is $\mathcal{O}(L^2)$. An important observation is that due to the scaling as $\mu = \gamma_4^2/L^4$, the dependence on the coordinates $\mathbf{n},\mathbf{m}$ in (20) can be neglected. Thus, a straightforward calculation yields

$$\langle\mathbf{1}|\mathbb{S}|\mathbf{1}\rangle = \frac{\sqrt{g}\omega_x\omega_y L^2}{2\sqrt{2}\gamma_4}. \tag{39}$$

One should observe that Eq. (39) is exactly the contribution of $\mathbf{k} = \mathbf{0}$ in the sum in (12). Physically, this means that in the ordered phase of the QSM for $g < g_c$ the leading behaviour of the eigenvalue of $\mathbb{S}_A$ associated with the flat vecto is simply obtained by isolating the term with $\mathbf{k} = \mathbf{0}$ in (12). This happens because of the "fast" decay as $\mu \propto 1/L^4$. This is not the case at the critical point [64], where $\mu \propto 1/L^2$. Moreover, this result suggests that one can decompose the correlator $\mathbb{S}$ as

$$\mathbb{S} = s_0|\mathbf{1}\rangle\langle\mathbf{1}| + \ldots, \quad \text{with } s_0 = \langle\mathbf{1}|\mathbb{S}|\mathbf{1}\rangle. \tag{40}$$

Here $s_0 \propto L^2$, and the dots are subleading terms that we neglect. By using (40) and the fact that $\mathbb{P}$ is finite in the limit $L \to \infty$, it is straightforward to show that the eigenvalue $e_1$ of $\mathbb{C}_A = \mathbb{P}_A \cdot \mathbb{S}_A$ in the limit $L \to \infty$ is given as (see [77] and [64])

$$e_1 = \langle\mathbf{1}|\mathbb{S}|\mathbf{1}\rangle\langle\mathbf{1}|\mathbb{P}|\mathbf{1}\rangle. \tag{41}$$

Here we have

$$\langle\mathbf{1}|\mathbb{P}|\mathbf{1}\rangle = \frac{1}{|A|} \sum_{\mathbf{n},\mathbf{m}\in A} \mathbb{P}_{\mathbf{n}\mathbf{m}}. \tag{42}$$

To proceed we now show that the expectation value $\langle\mathbf{1}|\mathbb{P}|\mathbf{1}\rangle$ decays as $\ln(L)/L$, i.e., with a multiplicative logarithmic correction. Note that the same scaling behaviour is observed at the critical point [64]. The derivation requires minimal modifications as compared with the critical case, and it is reported in A. The main ingredients are standard tools in the finite-size scaling theory, such as Poisson's summation formula and the Euler-Maclaurin formula.

Let us discuss the final result. Clearly, we can treat the contribution of the thermodynamic part (cf. (23)) and the finite-size part (cf. (24)) separately. Similar to what happens

at the critical point [64], the finite-size part contributes only if the boundary between the two subsystems is straight. For simplicity we consider the bipartition with $\omega_x = 1/p$ and $\omega_y = 1/q$, with $p, q \in \mathbb{N}$. Note that for $\omega_y < 1$ the boundary between the two subsystems is not straight, i.e., it has square corner. One obtains the generic thermodynamic contribution as

$$\langle \mathbf{1}|\mathbb{P}^{(th)}|\mathbf{1}\rangle =$$
$$\sum_{p'=0}^{p-1} \sum_{q'=0}^{q-1} \int_0^{1/p} dk_x \int_0^{1/q} dk_y \sin^2(\pi(k_x + p'/p)) \sin^2(\pi(k_y + q'/q)) \eta_{p',q'}(k_x, k_y). \quad (43)$$

The function $\eta_{p',q'}(k_x, k_y)$ reads as

$$\eta_{p',q'}(k_x, k_y) = \frac{4}{\pi^3 \sqrt{g}} \Big[ \frac{q}{(k_x + p'/p)^2} + \frac{p}{(k_y + q'/q)^2} + p\psi'(1 + k_y + q'/q)$$
$$+ \frac{q}{1 + k_x + p'/p} + \frac{q}{2(1 + k_x + p'/p)^2} \frac{q}{6(1 + k_x + p'/p)^3} + \dots \Big] \frac{\ln(L)}{L}. \quad (44)$$

The dots in the brackets denote terms with higher powers of $1/(k_x + p'/p)$. These can be derived systematically by using the Euler-Maclaurin formula. The function $\psi'(x)$ is the first derivative of the digamma function $\psi(x)$ with respect to $x$. The behaviour as $\ln(L)/L$ is clearly visible in (44). Similar to the critical point [64], $\eta_{p',q'}$ is determined by the low-energy part of the dispersion of the QSM. Finally, let us consider the finite-size contribution (24). From (24) it is clear that the finite-size correlator is regular for $\omega_y < 1$ and $\omega_x < 1$, whereas it exhibits a singularity for $\omega_y = 1$ or $\omega_x = 1$, i.e., for the case of straight boundary (see Fig 1 (b)). For the straight boundary this gives a contribution as $\ln(L)/L$, whereas it can be neglected if a corner is present. Again, this is exactly the same at the critical point [64]. The derivation of the singular contribution, which is present only for straight boundary, is reported in A.2. The final result reads

$$\langle \mathbf{1}|\mathbb{P}^{(L)}|\mathbf{1}\rangle = -\frac{1}{\sqrt{g}\pi} \frac{\ln(L)}{L}. \quad (45)$$

The minus sign in (45) implies that the presence of corners increases the prefactor of the logarithmic growth of $e_1$. After putting together Eq. (41), Eq. (39), Eq. (43) and Eq. (45), one obtains that

$$e_1 = \Omega' L \ln(L), \quad (46)$$

where the constant $\Omega'$ encodes information about the geometry of the bipartition and the model dispersion. In Eq. (46) we neglect subleading terms in the limit $L \to \infty$. From (46), after using (35) one obtains that

$$\delta\xi = \frac{\Omega}{\sqrt{L \ln(L)}}, \quad \text{with } \Omega = \frac{1}{\sqrt{\Omega'}}. \quad (47)$$

Few comments are in order. First, in the ordered phase $\delta\xi$ vanishes in the thermodynamic limit as a power law with $L$, except for a logarithmic correction. This is different at the critical point, where [64] $\delta\xi = \pi^2/\ln(L)$. The power-law decay of the entanglement gap in symmetry-broken phases has been also numerically observed in magnetic spin systems [34, 36] and in the ordered phase of the two-dimensional Bose Hubbard model [31]. Note, however, that even with state-of-the-art numerical methods it is challenging to observe the logarithmic correction. Finally, in Ref. [29] it has been suggested that in the presence

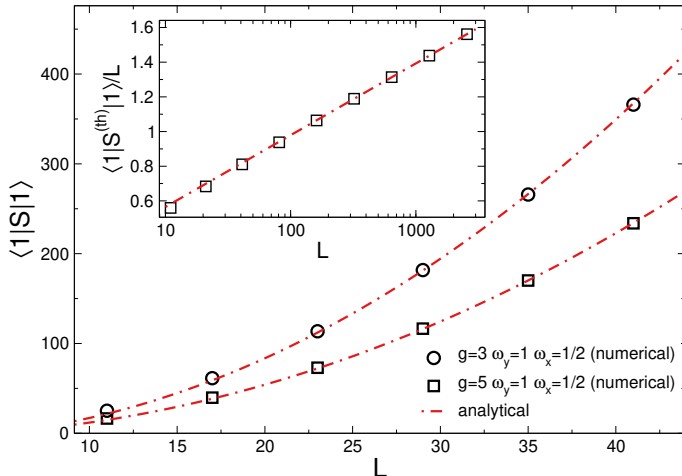

Figure 3: Flat-vector expectation value $\langle\mathbf{1}|\mathbb{S}|\mathbf{1}\rangle$ of the spin-spin correlator in the ordered phase of the QSM. The behaviour as $\langle\mathbf{1}|\mathbb{S}|\mathbf{1}\rangle \propto L^2$ is clearly visible. The dashed-dotted lines are the theory predictions (39). We show results for different aspect ratios $\omega_y, \omega_x$ (see Fig. 1) and quantum coupling $g$. The inset show the contribution of the thermodynamic part of the correlator $\mathbb{S}^{(th)}$ (cf. (18)) for $g = 5$. The behaviour as $\langle\mathbf{1}|\mathbb{S}^{(th)}|\mathbf{1}\rangle \propto L\ln(L)$ is clearly visible.

of continuous symmetry breaking the gaps in the lower part of the entanglement spectrum are

$$\delta\xi \propto (L\ln(L))^{-1}. \tag{48}$$

This different from (47) (note the square root in (47)). The unexpected square root in Eq. (47) could be explained by the way in which in the QSM the spherical constraint is enforced (cf. (16)). Further study would be needed to clarify this issue. Finally, it is interesting to understand the behaviour of $\delta\xi$ as the critical point is approached from the ordered side of the transition. A natural scenario is that upon approaching the transition the $1/\sqrt{L}$ is "gapped" out and it gives an extra $1/\sqrt{\ln(L)}$, which allows to recover the expected result [64] $\delta\xi \propto 1/\ln(L)$.

# 6   Numerical results

In this section we provide numerical evidence supporting the analytic result derived in section 5. Let us start discussing the finite-size scaling of the expectation value $\langle\mathbf{1}|\mathbb{S}|\mathbf{1}\rangle$. We report numerical data in Fig. 5, for fixed $g = 3$ (circles) and $g = 5$ (squares). We only show data for the bipartition with straight boundary $\omega_y = 1$

(see Figure 1 (a)) and for $\omega_x = 1/2$. The expected behaviour as $\langle\mathbf{1}|\mathbb{S}|\mathbf{1}\rangle \propto L^2$ is visible. The dashed-dotted line in the figure is the analytic result in Eq. (39), which is in perfect agreement with the numerical data. Again, we should stress that Eq. (39) originates only from the finite-size part $\mathbb{S}^{(L)}$ (cf. (18)). However, it is interesting to investigate the finite-size scaling of the flat-vector expectation value calculated using the thermodynamic contribution $\mathbb{S}^{(th)}$. We report this analysis in the inset of Fig. 5 plotting $\langle\mathbf{1}|\mathbb{S}^{(th)}|\mathbf{1}\rangle/L$ versus $L$. Data are for $g = 5$. Interestingly, the figure shows that $\langle\mathbf{1}|\mathbb{S}^{(th)}|\mathbf{1}\rangle \propto L\ln(L)$. This confirms that at the leading order in $L$ the expectation value $\langle\mathbf{1}|\mathbb{S}|\mathbf{1}\rangle$ is dominated by the contribution of the zero mode. Finally, we should mention that it would be interesting

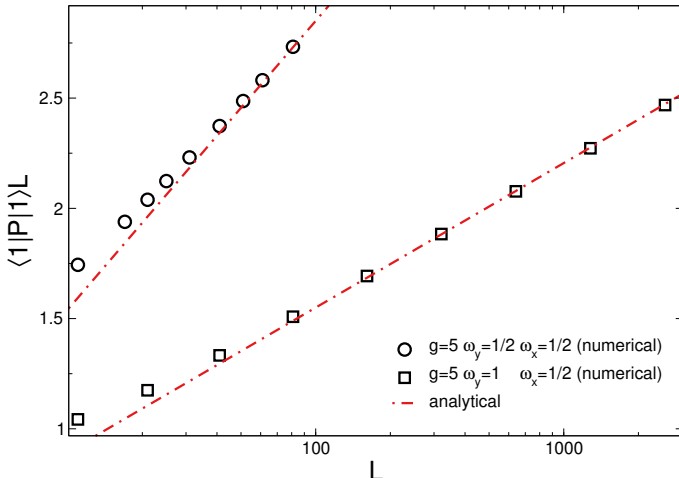

Figure 4: Rescaled flat-vector expctation value $\langle\mathbf{1}|\mathbb{P}|\mathbf{1}\rangle L$ of the momentum operator in the ordered phase of the QSM. We show data for several bipartitions with aspect ratios $\omega_x, \omega_y$ (see Fig. 1). For $\omega_y < 1$ the boundary between the two subsystems is not smooth (see Fig. 1 (b)). Symbols are exact numerical results. The dashed-dotted lines are analytic predictions from (43) and (45).

to clarify the origin of the logarithmic divergence of the thermodynamic contribution.

Let us now discuss the flat-vector expectation value of the momentum correlator $\langle\mathbf{1}|\mathbb{P}|\mathbf{1}\rangle$. In contrast with the spin correlator, both the thermodynamic and the finite-size part (cf. (22)) contribute to the leading behaviour at large $L$. Our numerical data are reported in Fig. 6. In the figure we plot $\langle\mathbf{1}|\mathbb{P}|\mathbf{1}\rangle L$ versus $L$. We show data for $\omega_x = 1/2$, $\omega_y = 1$ and $\omega_y = 1/2$. Note that for $\omega_y = 1$ the boundary between $A$ and its complement is straight. The numerical data in Fig. 6 confirm the expected behaviour as $\ln(L)/L$ in Eq. (43) and Eq. (45). For $\omega_y = 1$ the prefactor of the logarithm is obtained by summing Eq. (43) and Eq. (45), whereas in the presence of a square corner only Eq. (43) has to be considered. Finally, we discuss the largest eigenvalue $e_1$ of the restricted correlation matrix $\mathbb{C}_A$ (cf. (33)). The entanglement gap $\delta\xi$ is obtained from $e_1$ via Eq. (35). Note that the vanishing of $\delta\xi$ is reflected in a diverging $e_1$. We show numerical data for $e_1/L$ in Fig. 6 plotted versus $L$. We consider several aspect ratios $\omega_x$ and $\omega_y$, focusing on $g = 5$. In all the cases the data exhibit the behaviour $e_1 = \Omega' L \ln(L)$. The constant $\Omega'$, which depends on the geometry and on low-energy properties of the QSM is obtained by combining Eq. (41) with Eq. (39) (43) (45). The analytic predictions are reported in Fig. 6 as dashed-dotted lines and are in perfect agreement with the numerical data. This implies that the entanglement gap $\delta\xi$ satisfies (47).

# 7 Conclusions

We investigated the entanglement gap in the magnetically ordered phase of the two-dimensional QSM. Our main result is that the entanglement gap decays as $\delta\xi = \Omega/\sqrt{L \ln(L)}$. We analytically determined the constant $\Omega$, which depends on the geometry of the bipartition and on the low-energy physics of the model.

There are several intriguing directions for future work. First, it would be interesting to explore whether is possible to extend our results to the $N$-vector model at finite $N$. An interesting question is whether the discrepancy with the results of Ref. [29] can be

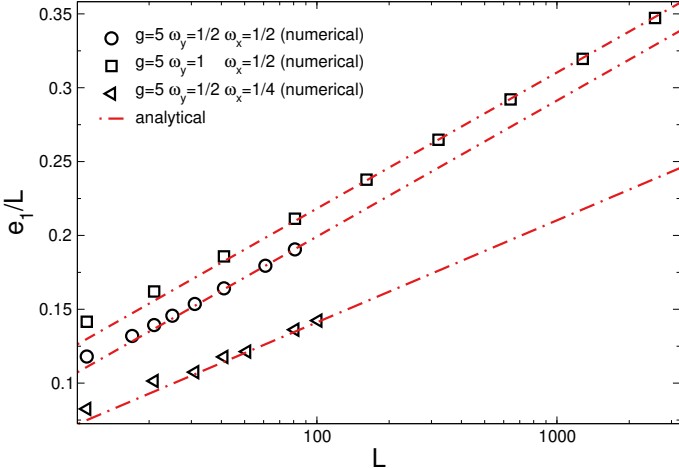

Figure 5: Largest eigenvalue $e_1$ of the correlation matrix $\mathbb{C}$ restricted to $A$. We plot $e_1/L$ versus $L$. Note the logarithmic scale on the $x$-axis. Symbols are exact numerical data. The dashed-dotted lines are analytic predictions. Note that for $\omega_y$ the boundary between the two subsystems has a corner (see Fig. 1 (b)).

attributed to the large $N$ limit. Furthermore, it is important to understand how the scaling of the entanglement gap depends on dimensionality. This issue could be easily addressed because the QSM is exactly solvable in any dimension. Another intriguing direction is to further investigate the role of corners. For instance, it would be interesting to investigate the dependence of the entanglement gap on the corner angle. It would be also interesting to investigate how the outlined scenario is affected by long-range interactions. This should be straightforward because the QSM is exactly solvable also in the presence of long-range interactions. An exciting possibility is to investigate what happens to the entanglement gap in the presence of disorder [78–81]. Finally, a very interesting direction is to study $\delta\xi$ after a quantum quench. This could be addressed using the results of Ref. [82].

## Acknowledgements

I would like to thank Sascha Wald and Raul Arias for several discussions in a related project. I acknowledge support from the European Research Council under ERC Advanced grant 743032 DYNAMINT.

## A    Derivation of the expectation value $\langle \mathbf{1}|\mathbb{P}|\mathbf{1}\rangle$

In this appendix we derive the large $L$ behaviour of the expectation value of the momentum correlator with the flat vector $\langle\mathbf{1}|\mathbb{P}|\mathbf{1}\rangle$ (cf. (42)). We consider the leading, i.e, the thermodynamic limit, as well as the first subleading contributions. The main goal is to show that the expectation value exhibits a multiplicative logarithmic correction. Two types of contributions are present. One originates from the thermodynamic limit of the correlator, whereas the second one is due to the first subleading. The latter is present only for straight boundary between the two subsystems (see Fig. 1).

## A.1 Thermodynamic contribution

Here derive the thermodynamic contribution, which is given as $\langle \mathbf{1}|\mathbb{P}^{(th)}|\mathbf{1}\rangle$. Here $|\mathbf{1}\rangle$ is the flat vector in region $A$, i.e,

$$|\mathbf{1}\rangle = \frac{1}{\sqrt{|A|}}(1,1,\ldots,1), \quad |A| = \ell_x \ell_y. \tag{49}$$

The momentum correlation reads

$$\mathbb{P}^{(th)}_{\boldsymbol{nm}} = \frac{1}{4\sqrt{2g}\pi^2} \int_{-\pi}^{\pi} d\boldsymbol{k}\, e^{i\boldsymbol{k}(\boldsymbol{n}-\boldsymbol{m})}\sqrt{\mu+\omega_{\boldsymbol{k}}}. \tag{50}$$

After performing the sum over $\boldsymbol{n}$ and $\boldsymbol{m}$ in (50), and after changing variables to $k_x' = L\omega_x k_x/\pi$ and $k_y' = L\omega_y k_y/\pi$, we obtain

$$\langle \mathbf{1}|\mathbb{P}^{(th)}|\mathbf{1}\rangle = \frac{2\sqrt{2}}{\sqrt{g}L^4\omega_x^2\omega_y^2} \int_0^{L\omega_x/2} dk_x \int_0^{L\omega_y/2} dk_y \frac{\sin^2(\pi k_x)\sin^2(\pi k_y)}{\sin^2\left(\frac{\pi}{L\omega_x}k_x\right)\sin^2\left(\frac{\pi}{L\omega_y}k_y\right)}$$
$$\times \left[\mu + 2 - \cos\left(\frac{2\pi}{L\omega_x}k_x\right) - \cos\left(\frac{2\pi}{L\omega_y}k_y\right)\right]^{\frac{1}{2}}. \tag{51}$$

To extract the large $L$ behaviour of (51) it is useful to split the integration domains $[0, L\omega_x/2]$ and $[0, L\omega_y/2]$ to write

$$\langle \mathbf{1}|\mathbb{P}^{(th)}|\mathbf{1}\rangle = \frac{2\sqrt{2}}{\sqrt{g}L^4\omega_x^2\omega_y^2} \sum_{l_x=0}^{L/2-1}\sum_{l_y=0}^{L/2-1} \int_{l_x\omega_x}^{(l_x+1)\omega_x} dk_x \int_{l_y\omega_y}^{(l_y+1)\omega_y} dk_y$$
$$\times \frac{\sin^2(\pi k_x)\sin^2(\pi k_y)}{\sin^2\left(\frac{\pi}{L\omega_x}k_x\right)\sin^2\left(\frac{\pi}{L\omega_y}k_y\right)}\left[\mu + 2 - \cos\left(\frac{2\pi}{L\omega_x}k_x\right) - \cos\left(\frac{2\pi}{L\omega_y}k_y\right)\right]^{\frac{1}{2}}. \tag{52}$$

We now restrict ourselves to the case with $\omega_x = 1/p$ and $\omega_y = 1/q$, with $p, q$ positive integers. After a simple shift of the integration variables as $k_x \to k_x - l_x\omega_x$ and $k_y \to k_y - l_y\omega_y$, one obtains

$$\langle \mathbf{1}|\mathbb{P}^{(th)}|\mathbf{1}\rangle = \frac{2\sqrt{2}p^2q^2}{\sqrt{g}L^4} \sum_{p'=0}^{p-1}\sum_{q'=0}^{q-1}\sum_{l_x=0}^{L/(2p)-1}\sum_{l_y=0}^{L/(2q)-1} \int_0^{1/p} dk_x \int_0^{1/q} dk_y$$
$$\times \frac{\sin^2(\pi(k_x+l_x+p'/p))\sin^2(\pi(k_y+l_y+q'/q))}{\sin^2\left(\frac{p\pi}{L}(k_x+l_x+p'/p)\right)\sin^2\left(\frac{q\pi}{L}(k_y+l_y+q'/q)\right)}$$
$$\times \left[\mu + 2 - \cos\left(\frac{2p\pi}{L}(k_x+l_x+p'/p)\right) - \cos\left(\frac{2q\pi}{L}(k_y+l_y+q'/q)\right)\right]^{\frac{1}{2}}. \tag{53}$$

We now focus on the behaviour at $g < g_c$. We set $\mu = \gamma_4/L^4$ (cf. (31)), and we expand (53) in the limit $L \to \infty$. This gives

$$\langle \mathbf{1}|\mathbb{P}^{(th)}|\mathbf{1}\rangle =$$
$$\frac{4}{\sqrt{g}\pi^3 L} \sum_{p'=0}^{p-1}\sum_{q'=0}^{q-1}\sum_{l_x=0}^{L/(2p)-1}\sum_{l_y=0}^{L/(2q)-1} \int_0^{1/p} dk_x \int_0^{1/q} dk_y \frac{\sin^2(\pi(k_x+p'/p))\sin^2(\pi(k_y+q'/q))}{(k_x+l_x+p'/p)^2(k_y+l_y+q'/q)^2}$$
$$\times \left[\frac{\gamma_4}{2\pi^2 L^2} + p^2(k_x+l_x+p'/p)^2 + q^2(k_y+l_y+q'/q)^2\right]^{\frac{1}{2}}. \tag{54}$$

Here we used the periodicity of the trigonometric functions. The term $\gamma_4/L^2$ can be neglected for $L \to \infty$. Importantly, as a result of the large $L$ limit, Eq. (54) depends only on the low-energy part of the dispersion of the QSM, although it contains non-universal information. We now have to determine the asymptotic behaviour of the sum over $l_x, l_y$ in (54), i.e., of the function $\eta_{p',q'}(k_x, k_y)$ defined as

$$\eta_{p',q'}(k_x, k_y) = \frac{4}{\sqrt{g}\pi^3 L} \sum_{l_x=0}^{L/(2p)-1} \sum_{l_y=0}^{L/(2q)-1} \frac{[p^2(k_x + l_x + p'/p)^2 + q^2(k_y + l_y + q'/q)^2]^{\frac{1}{2}}}{(k_x + l_x + p'/p)^2(k_y + l_y + q'/q)^2}. \quad (55)$$

The asymptotic behaviour of $\eta$ in the limit $L \to \infty$ can be obtained by using the Euler-Maclaurin formula. Given a function $f(x)$ this is stated as

$$\sum_{x=x_1}^{x_2} f(x) = \int_{x_1}^{x_2} f(x)dx + \frac{f(x_1) + f(x_2)}{2} + \frac{1}{6}\frac{f'(x_2) - f'(x_1)}{2!} + \dots \quad (56)$$

In (56) the dots denote terms with higher derivatives of $f(x)$ calculated at the integration boundaries $x_1$ and $x_2$. These can be derived to arbitrary order. To proceed, we first isolate the term with either $l_x = 0$ or $l_y = 0$ in (55). The remaining sum after fixing $l_x = 0$ or $l_y = 0$ can be treated with (56). We define this contribution to the large $L$ behaviour of $\eta_{p',q'}$ as $\eta_0$, which is given as

$$\eta_0 = \frac{4}{\sqrt{g}\pi^3}\left[\frac{q}{(k_x + p'/p)^2} + \frac{p}{(k_y + q'/q)^2}\right]\frac{\ln(L)}{L}. \quad (57)$$

In the derivation of (57) we neglected the boundary terms in (56) because they are subleading.

We are now left with the sums over $l_x \in [1, L/(2p)]$ and $l_y \in [1, L/(2q)]$ in (55). These be calculated again by using (56). We first apply (56) to the sum over $l_x$. We have two contributions. The first one is obtained after evaluating the integral in (56) at $x_2 = L/(2p)$. After expanding the result for $L \to \infty$, we obtain the contribution $\eta_1$ given as

$$\eta_1 = \sum_{l_y=1}^{L/(2q)} \frac{4p}{\sqrt{g}\pi^3(k_y + l_y + q'/q)^2}\frac{\ln(L)}{L}. \quad (58)$$

Note the term $\ln(L)/L$ in (58). The sum over $l_y$ in (58) can be performed exactly to obtain in the large $L$ limit

$$\eta_1 = \frac{4}{\sqrt{g}\pi^3}p\psi'(1 + k_y + q'/q)\frac{\ln(L)}{L}. \quad (59)$$

Here $\psi'(z)$ is the first derivative of the digamma function $\psi(z)$ with respect to its argument. The second contribution is obtained by evaluating the integral in (56) at $x_1 = 1$. The remaining sum over $l_y$ cannot be evaluated analytically. However, one can, again, treat the sum over $l_y$ with (56). After neglecting the boundary terms in (56), which are subleading for large $L$, and after evaluating the integral in (56) at $x_2 = L/(2q)$, we obtain the contribution $\eta_2$ as

$$\eta_2 = \frac{4}{\sqrt{g}\pi^3}\frac{q}{1 + k_x + p'/p}\frac{\ln(L)}{L}. \quad (60)$$

Having discussed the contribution which derives from approximating the sum over $l_x$ in (55) with the integral in (56), we finally focus on effect of the boundary terms in (56). Let us consider the first boundary term (first term in the second row in (56)). A term

as $\ln(L)/L$ is obtained by fixing $l_x = 1$, other contributions being subleading. After performing the sum over $l_y$ one obtains the first boundary contribution $\eta_{b1}$ as

$$\eta_{b1} = \frac{2}{\sqrt{g}\pi^3} \frac{q}{(1 + k_x + p'/p)^2} \frac{\ln(L)}{L}. \tag{61}$$

Similarly, the second boundary term (last term in (56)) gives

$$\eta_{b2} = \frac{2}{3\sqrt{g}\pi^3} \frac{q}{(1 + k_x + p'/p)^3} \frac{\ln(L)}{L}. \tag{62}$$

Note that boundary terms in (56) are expected to be small. Specifically, the $k$-th term is suppressed as $1/(k + 1)!$. The final result for $\eta(k_x, k_y, p, p', q, q')$ is obtained by putting together (57)(59) (60)(61)(62) to obtain

$$\eta_{p',q'}(k_x, k_y) = \eta_0 + \eta_1 + \eta_2 + \eta_{b1} + \eta_{b2}. \tag{63}$$

## A.2 Finite-size contribution

In this section we derive the leading behaviour in the large $L$ limit of $\langle \mathbf{1} | \mathbb{P}^{(L)} | \mathbf{1} \rangle$. Interestingly, we show that in the presence of a straight boundary between the two subsystems (see Fig. 1) one has the behaviour $\langle \mathbf{1} | \mathbb{P}^{(L)} | \mathbf{1} \rangle \propto \ln(L)/L$. On the other hand, in the presence of corners the multiplicative logarithmic correction is absent. The finite-size correlator reads as (cf. (24))

$$\mathbb{P}^{(L)}_{\boldsymbol{nm}} = -\frac{1}{4\sqrt{g}\pi} \sum_{l,l'=-\infty}^{\infty}{}' e^{-\sqrt{2\mu}\sqrt{(lL+n_x-m_x)^2+(l'L+n_y-m_y)^2}}$$

$$\times \left[ \frac{1}{[(lL + n_x - m_x)^2 + (l'L + n_y - m_y)^2]^{3/2}} + \frac{\sqrt{2\mu}}{(lL + n_x - m_x)^2 + (l'L + n_y - m_y)^2} \right]. \tag{64}$$

Crucially, if $\omega_x < 1$ and $\omega_y < 1$, the denominators in (64) are never singular. This implies that the logarithmic correction is not present, which can be straightforwardly checked numerically. Let us now consider the situation with $\omega_x < 1$ and $\omega_y = 1$. Now, a singularity appears in the limit $L \to \infty$ for $l = 0$ and $l' = \pm 1$. We numerically observe that only the first term in (64) gives rise to a singular behaviour. Thus, we neglect the second term and fix $l = 0$, obtaining

$$\langle \mathbf{1} | \mathbb{P}^{(L)} | \mathbf{1} \rangle = -\frac{1}{4\sqrt{g}\pi L^2 \omega_x} \sum_{l'=-\infty}^{\infty}{}' \sum_{n_x,m_x=0}^{L\omega_x} \sum_{n_y,m_y=0}^{L-1} \frac{e^{-\sqrt{2\mu}\sqrt{(n_x-m_x)^2+(l'L+n_y-m_y)^2}}}{((n_x - m_x)^2 + (l'L + n_y - m_y)^2)^{3/2}} \tag{65}$$

Only the differences $n_x - m_x$ and $n_y - m_y$ appear in (65). Thus, it is convenient to change variables to $x = n_x - m_x$ and $y = n_y - m_y$, to obtain

$$\langle \mathbf{1} | \mathbb{P}^{(L)} | \mathbf{1} \rangle = -\frac{1}{4\sqrt{g}\pi L^2 \omega_x} \sum_{l'=-\infty}^{\infty}{}' \sum_{x=-L\omega_x}^{L\omega_x} \sum_{y=-(L-1)}^{L-1}$$

$$\left( L\omega_x + 1 - |x| \right)(L - |y|) \frac{e^{-\sqrt{2\mu}\sqrt{x^2+(l'L+y)^2}}}{(x^2 + (l'L + y)^2)^{3/2}}. \tag{66}$$

Again, the singular behaviour occurs for $x \approx 0$ and $y \approx -lL$, with $l' = \pm 1$. In this limit we can neglect the exponential in (67) because it is regular. Thus, we obtain

$$\langle \mathbf{1} | \mathbb{P}^{(L)} | \mathbf{1} \rangle = -\frac{1}{4\sqrt{g}\pi L^2 \omega_x} \sum_{l'=-\infty}^{\infty}{}' \sum_{x=-L\omega_x}^{L\omega_x} \sum_{y=-(L-1)}^{L-1} \frac{(L\omega_x + 1 - |x|)(L - |y|)}{(x^2 + (l'L + y)^2)^{3/2}}. \tag{67}$$

To proceed, let us now consider the case with $l = 1$. It is clear that the contribution from $l = -1$ is the same. We can restrict the sum over $x$ in (67) to $x > 0$ because of the symmetry $x \to -x$. We also restrict to $y < 0$ because the singularity in (67) occurs for $y < 0$. We now have

$$\langle \mathbf{1} | \mathbb{P}^{(L)} | \mathbf{1} \rangle = \frac{1}{2\sqrt{g}\pi L^2 \omega_x} \sum_{x=0}^{L\omega_x} \sum_{y=0}^{L-1} \frac{(L\omega_x + 1 - x)(y - L)}{(x^2 + (L - y)^2)^{3/2}}. \tag{68}$$

Now the strategy is to treat the sum (68) by using the Euler-Maclaurin formula (56). For instance, one can first apply (56) to the sum over $x$. One obtains that the leading term in the large $L$ limit is obtained by evaluating the integral in (56) at $\omega_x L$. One can also verify that the boundary terms in (56) can be neglected. A straightforward calculation gives

$$\langle \mathbf{1} | \mathbb{P}^{(L)} | \mathbf{1} \rangle = -\frac{1}{\sqrt{g}\pi} \frac{\ln(L)}{L}, \tag{69}$$

where the contribution of $l = -1$ in (67) has been taken into account.

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
