# Peer review of "Entanglement gap, corners, and symmetry breaking"

_SciPost Physics Core_

## Round 1 · Referee Report · Anonymous (Referee 1) · 2020-11-23

Report
In this paper the author studies the finite-size scaling of the lowest entanglement gap in the ordered phase of the 2D quantum spherical model n a square lattice which is exactly solvable. The main result of the paper is that asymptotically the gap decays as \Omega / \sqrt{L \log L} as opposed to the ~1/sqrt{\log L} scaling at the critical point, and the author also determines the corner contribution to \Omega.
In recent years the study of the entanglement properties in quantum many-body systems has become a central topic. This works provides an interesting and valuable contribution to the field, and builds also an interesting basis for future works. The paper is written in a clear way, and the results have been carefully checked also numerically.
For these reasons I can recommend this paper to be accepted in SciPost Physics. I have only minor comments, listed below, which the author may want to consider when revising the manuscript.
Comments and questions:
1) It would be interesting to extend the discussion why a different scaling behavior was predicted in Ref.[29], in order to better understand the difference with the result in this paper. A brief comment is made around Eq.(48) and in the conclusions, but if there is any more information that could be added here, that would be useful (e.g. based on what approach/assumptions the result in Ref.[29] was obtained).
2) In the conclusions the author mention the importance to understand how the scaling of the entanglement gap depends on dimensionality and the range of interactions. I was wondering if there is "natural" conjecture that could be made here, or if it is completely unknown.
3) Here is a list of minor typos I spotted while reading: - p3 first line of Sec.2: "cubic lattice" -> "square lattice" (since it is 2D) - Eq.(9) A point "." is missing - p4, 4th line after Eq (17): remove either "finite" or "nonzero". Same line: "at the critical point" - p8, 3d line after Eq (39): "vecto" -> "vector" - p10, Sec 6, there is an extra new line before "(see Figure 1(a))" which should be removed. - p14, 3 lines after Eq (57): "These be calculated" -> "These can be calculated" - please check the arxiv references in the bibliography, especially the links which do not seem to work properly (e.g. [29], [39]).

---

## Editorial Decision

unknown